# Migrant Women in Shantytowns in Southern Spain: A Qualitative Study

**DOI:** 10.3390/ijerph20085524

**Published:** 2023-04-14

**Authors:** Fernando Jesús Plaza del Pino, Lucía Muñoz Lucena, Nadia Azougagh, Ana Gómez Haro, Belén Álvarez Puga, Silvia Navarro-Prado, María Jesús Cabezón-Fernández

**Affiliations:** 1Department of Nursing Physiotherapy and Medicine, University of Almería, 04120 Almería, Spain; 2Centre for Migration Studies and Intercultural Relations, University of Almería, 04120 Almería, Spain; 3Entrefronteras Social Producer, 29010 Málaga, Spain; 4Organization Alianza por La Solidaridad-Actionaid, 41018 Sevilla, Spain; 5Distrito Sanitario Poniente de Almería, Servicio Andaluz de Salud, 04700 Almería, Spain; 6Department of Nursing, Faculty of Health Sciences, Melilla Campus, University of Granada, 52005 Melilla, Spain; 7Department of Geography, History and Humanities, University of Almería, 04120 Almería, Spain

**Keywords:** migrants, shantytown, Spain, social and labor inequalities, women

## Abstract

The increase in intensive agriculture in Southern Spain, and the increasing need for migrant women, has led to the appearance of numerous shantytowns alongside greenhouses. In the last few years, the number of women who live in them has increased. This qualitative study delves into the experiences and future expectations of migrant women who live in shantytowns. Thirteen women who live in shantytowns in Southern Spain were interviewed. Results: Four themes emerged: dreams vs. reality, life in the settlements, worse for women, and “the papers”. Discussion and Conclusions. Priority should be given to the care of women who live in shantytowns with specific programs; society must work to end these shantytowns and facilitate agricultural workers with access to housing; it is necessary to allow the resident registration of the people who live in shantytowns.

## 1. Introduction

Since the 1990s, immigration has become a phenomenon of great demographic and economic importance. The shift of societies from an industrial model to a post-industrial one has brought with it changes in migration trajectories, the management of migration fluxes, and working conditions [1]. The growth of the agro-industry and these new migratory movements to Southern Europe have resulted in Italy, Greece, Portugal, and Spain becoming the main exporters of horticultural crops in Europe. In Southern Spain, the agricultural sector began to change its production model towards a model of intensive agriculture in greenhouses, or “under plastic”, in the 1980s [2]. The provinces of Almeria, with more than 32,000 hectares [3], and Huelva, with 11,700 hectares, are dedicated to the production of red berries [4] and have become large horticultural product producers focused on the international market [2,3].

In the sector of intensive agriculture, the globalization of production and markets, together with a strong division of labor, has led to a change from the old model [2] into one that includes seasonality and job insecurity and is characterized by flexibility in hiring, precarious wages, and a lack of worker’s rights [5,6], making it unattractive for the autochthonous population but attractive for the migrant population [7]. Agricultural work is, for many migrants, the first access point to the Spanish labor market. In both Almeria and Huelva, the availability of workers for intensive agriculture depends on the combination of legal and illegal migrants [5], as the restrictive immigration policies have not avoided the irregular hiring or arrival of foreigners without legal papers [8]. These characteristics of the sector, determined by its seasonality, make difficult the access to basic rights, and reduce, for example, the lack of access to unemployment benefits, due to the low registration in Social Security among migrant workers, as well as healthcare rights and social services [9].

These characteristics of the agricultural sector in the south of Spain lead to migrant workers establishing their residences in areas near the greenhouses via the creation of informal or shantytown neighborhoods, called settlements, that vary in size and in the nationalities of the inhabitants. The origin of the irregular or informal settlements is directly associated with the growth of the agrarian economy, the intensification of the crops, and the need for labor [10]. This sub-standard housing is built from discarded material, such as pallets, cardboard, plastic, and blankets, and without electricity or running water. The informal supply of electricity and/or the use of fires to heat the shacks have resulted in fires in the settlements on many occasions [10,11]. Furthermore, over the years, it is not only the number of settlements that has increased, but also the volume of workers living in them permanently, which is partly due to the expansion of crops that have prolonged the months of harvest work [12]. Addressing the magnitude of this phenomenon and the social situation of the people who live in these settlements is very complicated as there are no current studies, or official statistical sources, that can provide data [13]. It is estimated, however, that about 7000 people live in informal settlements in Almeria and about 5000 live in informal settlements in Huelva [1].

Irregular migration is a common and visible characteristic of global movement [14]. For irregular workers, work and life instability and vulnerability is greater, as they must face problems such as a lack of housing, a lack of legal protection, and police control and persecution [15,16]. The main objective of migrant individuals in an irregular situation in Spain is to obtain “the papers” to legalize their administrative situation. The most frequent and feasible regularization mechanism is so-called “social ties” [17]. The foreigner can obtain a residency permit due to exceptional circumstances if he or she is able to justify living in Spain for three consecutive years and can demonstrate this with a residency registration, a job offer, a certification of social integration, or some type of family tie in the country. However, living in a shantytown makes accessing the residency register impossible [11].

In the provinces of Huelva and Almeria, intensive agriculture has increased the stratification of the labor market by origin and gender [18]. Presently, women are employed in red berry harvesting tasks (Huelva) or the canning of horticultural products in canning warehouses (Almeria), while men perform planting and harvesting tasks (Almeria/Huelva). This phenomenon has changed over time, since at the start of intensive agriculture only migrant men were employed, following the model of the breadwinner man [19].

The women migrants in the agricultural sector in the South of Spain are a recent addition, following the footsteps of the general trend of the feminization of the migrations [7] motivated by the phenomenon of the trans-nationalization of the global chain of care, which has promoted the migrations of the “leaders of the home” [19], as well as other migration dynamics of women who decide to embark in a solitary migration project [20]. In the particular case of Huelva, Gualda [7] points to a “triple relay” of the agricultural worker: male migrant workers have replaced native workers, and, more recently, women from the East, and more recently, women from Africa. Migrant women are exposed to greater risks than men with respect to discrimination, exploitation, and violence [21,22]. Along with the migrant movements of women, other aspects that have not been well studied in previous decades have been incorporated, such as gender-based violence, the changes in gender roles in migrant families, the health of women in migrant trajectories, the phenomena of prostitution and human trafficking, etc. Gender has shifted from being a variable of analysis within migration studies to becoming a cross-sectional variable, resulting in a change in perspective in the study of migration phenomena [23]. A greater and more detailed research activity with respect to the situation of migrant women is needed [24].

In Huelva, the main path to accessing the labor market in intensive agriculture is the recruitment in the country of origin (Morocco) [25] of women with a specific profile: women with children at their care who will have to return to their country once the season is finished [2,11,26]. In addition to migrant women’s primary access to employment in agriculture being via their country of origin [7], and employment as a means of family reunification, we must also consider other internal migratory dynamics of international migrants, such as obtaining year-round employment by working between regions during different harvesting seasons or abandoning settlements to avoid traumatic events experienced due to phenomena associated with abuse, which occurred in Huelva in 2018 [11,27]. These are phenomena that have resulted in a particularly female migratory circuit.

In this study, we delve into female migration among Moroccan individuals within the context of intensive agriculture in Huelva and Almeria, as described by the women themselves, examining the migratory process, the experience of living in shantytown settlements, and the women’s expectations for the future.

## 2. Materials and Methods

A qualitative study was designed, which followed an interpretative phenomenological approach as the best strategy for the understanding of human experiences [28]. Instead of studying an event, we considered the perspective of those who experienced it, as people try to create meaning from their experiences [29]. In our case, we focused on the experiences of a group of women with specific living conditions: migrant women who live in a shantytown in Southern Spain.

The study informants were migrant women, hereafter MW, who lived in the shantytown settlements in Southern Spain, more specifically, in the provinces of Almeria and Huelva. An intentional sampling method was utilized and concerned a selection of participants from different age groups, different years of stay in Spain, and different levels of knowledge of Spanish to ensure a broad coverage of the specific environment. The research team recruited the participants via non-governmental organizations that worked in the shantytowns, and which provided us with the contact information of women who met the inclusion criteria: being migrants and living in a shantytown. Participation was voluntary. Interviews were carried out until the research team decided that saturation of data had been reached [30], after which the data collection ended. 

The total number of participants in the study was 13. The characteristics of the informants are shown in Table 1.

The mean age of the informants was 33.3 years old. Except for four women, all of them had at least one child. The country of origin of all the informants was Morocco. In general, they did not speak Spanish, and, except for two women, all of them were legally irregular in Spain. Almost all of them had arrived in Spain between 2017 and 2019.

The data collection was performed via semi-structured interviews following a guide of a set of open-ended questions to ease the in-depth discussion of the subjects of interest.

The interviews were conducted during the months of January and March 2022. The interviews were conducted in the surroundings of the shantytown. In some cases, they were conducted in the shack of the informant and, in others, in open spaces, depending on their preference, to establish an adequate environment to facilitate the expression of feelings and emotions in an atmosphere of sincerity. The mean duration of the interviews was fifty minutes, and they were conducted in Arabic by a researcher of Moroccan origin (N.A.). The interviews were recorded with the permission of the participants.

The interviews were transcribed into Arabic, and, afterwards, they were translated into Spanish by the Moroccan researcher herself. The data were collected, managed, classified, and organized with the help of the qualitative data analysis software ATLAS-ti 8.0.

Firstly, an iterative reading of the transcripts was performed. The themes initially identified were aligned to the three open-ended questions formulated in each semi-structured interview. These questions were related to the expectations of the participants about their migratory project, their living conditions, and their expectations for the future. It should be noted that the theoretical framework to build these questions was based on the previous scientific literature and the objectives established in our study. Subsequently, the categories and their corresponding subcategories were identified according to the themes that emerged [31]. The research team was rigorous during the entire process.

The reliability of the data analysis was verified by comparing the codification of the three researchers (F.J.P.d.P., S.N.-P., and M.J.C.-F.) who codified the transcriptions independently. Afterwards, the consistencies and discrepancies between the researchers were determined. The disagreements were resolved by consensus. The research team constantly reflected during the entire process of analysis. The citations that best represented the categories were extracted and presented in the results section. The COREQ criteria were utilized as a guide to provide information about this qualitative study [32].

The research protocol was approved by the Research Ethics Committee from the Department of Nursing, Physical Therapy, and Medicine at the University of Almería (protocol number; EFM 171/2022). To guarantee the anonymity and confidentiality, a code was assigned to each participant. In every case, a written consent form and an explanation was provided to each of the MW.

## 3. Results

After the initial identification of the 52 codes in the data, more important codes appeared that were grouped into categories. A more in-depth analysis identified four thematic main categories (Table 2).

The four main categories, which arose after the codification, and their corresponding sub-categories, supported by the MW narratives, are described in detail below.

### 3.1. Dreams and the Migrant Reality

#### 3.1.1. Broken Dreams

The ideas of these women in their country of origin about the possible improvements in their lives and the lives of their families following migration, contrasted with what they experienced after their arrival in Spain, are presented as expressed by our informants.
*I saw that the people who came from Spain came with money, came back with a car, with many things, the people did not talk about their suffering there. I didn’t think about anything, only that I wanted to change my life*. MW4
*I always thought that Spain had everything, and that everything would be easier, but that’s not the case. I have suffered a lot and with many things, to look for work, to look for papers, for everything. Things are not easy here*. MW9

#### 3.1.2. Reality

Many of our informants had arrived in Spain with a contract from their country of origin to work during the red berry season in Huelva, which included country entry and exit dates, but decided to stay in Spain. Leaving these agricultural exploitations implies becoming illegal in Spain.
*I came here from Morocco with a contract to work during the strawberry season and stayed, I did not want to go back*. MW4
*We came with a work contract at origin. We thought that we would come here and improve our lives that we could find work, and things have not worked out as we expected*. MW12
*I thought there would be a lot of work with or without legal documents. In the end, I’m here without work and living in a shantytown*. MW13

The companies that wrote the seasonal worker contracts in Morocco only selected women with family loads to avoid their remaining in Spain at the end of the season.
*You can’t come to Huelva without being married or without children; young people cannot come during the season to work in Huelva*. MW5

Our MW explained to us that the work conditions suggested in the contracts they signed in Morocco were not met by the agricultural business owners.
*I worked in Huelva from 8 in the morning to 7 in the evening (…) it’s not supposed to be that I bring you here to work and you work…no, they exploit us*. MW2
*They only pay around 35 Euros, because they subtract transport, the boss explained to us, “look this what you earn in a day, but when you subtract…” we told him that they paid us less, and he said that we had to subtract rent, electricity, water, and transport*. MW6

### 3.2. Life in the Settlement

#### 3.2.1. Arriving at the Settlement

The situation of irregular work makes it so that they cannot sign rent contracts, and on some occasions, they rent rooms in homes from fellow countrymen, but the lack of stable resources impedes them from keeping the rentals.
*First I rented a house in the neighborhood, I was there for a few months, but since I didn’t have work, I spent my money and could not pay, so I came to the settlement.*MW1
*I live here because I don’t have work and I don’t have any other means. I can’t go back to Morocco. Without work, no one lets us rent a house*. MW4

The impossibility of accessing rental housing was also experienced by MW with residency permits.
*I would love to be able to rent, but it’s impossible, I’ve searched but nothing*. MW1
*Here I’ve never found (a place) to rent, if I could find rent, I would not stay here, dear, I would rent, you get a house and rent, and share it with two or more, you help them with rent, and you live in the town*. MW3

#### 3.2.2. Women in the Settlement

The increase in the number of MW in the shantytowns is novel; until a few years ago, it was difficult to find a woman living in them.
*Three or four years ago, many women came, and since then, there are many women*. MW1
*The women worked in Huelva, when the season is over, they come here, and since they can’t pay for rent, they arrive to the settlements to build shacks, and live in peace, before living on the street*. MW5

#### 3.2.3. Living Conditions

The living conditions in the shantytown are difficult due to the materials used to make the shacks as well as the unhealthy surroundings and the difficulties experienced concerning the accessibility of potable water and electricity.
*This house is built with pallets, cardboard, plastic, and sheets inside, and nails, I built it with my own hands.*MW4
*Living under the plastic when it’s hot, you can’t stay inside, and when it’s cold, it’s too cold. There is no water, days with electricity, and days without it.*MW7
*People work under the plastic and live in the plastic, this is a problem, health problems and many things, but there is no other way*. MW5
*I, for example…haven’t had power for 4 days, although you buy food, everything goes bad, we people suffer.*MW3
*We have to bring water from very far and we cannot drink it, it’s only for washing*. MW8

#### 3.2.4. The Frequent Fires in Shantytowns

The materials used to build the shacks, as well as the electrical connections and the methods used to stay warm, lead to frequent accidents, especially during the winter. The residents lose their few belongings during the fires, as the informants have told us.
*We ran from the fire with whatever we had on, without clothes, we left it all inside. I came close to dying*. MW9
*All the Moroccan women who live in this area have lost their shack, all of our things*. MW11
*…I left everything, and one would tell me “take, take everything you care about”, and I said I don’t care about anything, I went out to the street, and that’s it…*MW3

All the women stated that the worst thing that could happen to them in the fires is that their “papers” are burned. These papers are the proof of their stay in Spain, and they accumulate them to request legal residency on the basis of social ties.
*I heard people screaming, and everyone began to leave, I didn’t pick anything up, I got my children and went out running, I didn’t get any clothes, only the backpack with the documents (…) you leave everything, the documents are the most important thing, the shack can burn, but not the documents*. MW1
*It’s been 3 years* (she can ask for residency), *and everything burns, I give them proof, but I don’t have anything…before the fire, at least I had some hope, but now…I don’t have anything…(cries)*. MW2
*I don’t have my papers here, I have them somewhere else. Here I only have clothes that I can grab when I’m running out*. MW4

The fear of fires is permanent, which leads to uncertainty and sleep problems.
*…and one doesn’t rest…that fear…*MW1
*Me since the fire, since that day I went out to the street, I ask God that nothing happens to me, and when I sleep, I ask God that I wake up well, we are not calm, we don’t sleep calmly, we sleep and think if I sleep…everything can burn, always sleeping with a lot of fear*. MW3

### 3.3. Worse for Women

The difficulties in the shantytown are worse for MW, as many of them have children due to the greater difficulties in finding work and the danger of suffering harassment.

#### 3.3.1. Children

On many occasions, the children had remained in Morocco, and many years pass until they are reunited.
*My son is very little, he stayed with my mother in Morocco* (her eyes fill with tears), *my son…my son*. MW4

The children who were in Spain were there because they were either born there or because their mothers were able to bring them after obtaining the residency papers. In any case, having the children with them, and being alone, was yet another difficulty for the mothers in terms of finding work.
*I only stay here…I would love to go to Huelva, because there is a lot of work right now, in Huelva, in Murcia, Granada…and having papers…but of course, my children have school here. How can I do it? There is no way…I would have to take them, but they would need school, housing, everything*… MW1
*In my case, I don’t move because my children are small*. MW9

#### 3.3.2. More Difficulties for Working

For women in an irregular situation, the most accessible work is in the greenhouses, but the farmers in the area consider the women less able to complete the work.

Many MW indicated that farmers had offered work in exchange for sexual favors and described other situations of sexual harassment that they had suffered.
*They know we have to work, the boss told me that if I was with him, I could work as much as I wanted*. MW8
*They tell us whatever they want, we have to deal with it, they know we are alone and need the money*. MW12

Being pregnant, together with the lack of knowledge about their rights, results in cases from the women such as the one described here by our informant who was later fired following her pregnancy.
*I was working with one for 9–10 years…since I was a “harraga”* (an immigrant without papers in Arabic) *but since he found out I was pregnant, well, when I told him that he had to deal with the maternity leave paperwork, the insurance, and all of that, he didn’t like the idea, and fired me…I didn’t know about my rights*. MW1

#### 3.3.3. Without Resources

The lack of income results in many of the women experiencing very harsh situations.
*On many occasions, we even have to go to the dumpsters to look for clothes and things we don’t have, because of necessity*. MW13

This extreme situation of necessity has forced some of them to become prostitutes.
*There are many girls who come, and when they see the situation of exploitation, they go to the streets, I’m one of them, I’m not lying…you become a prostitute because you don’t have any other choice…I’ve found myself without being able to eat…I myself sell my breath, it’s hard…why? Because of the situation I’m in*… MW2

#### 3.3.4. Fear of Harassment

The vulnerability of these women, who are alone and/or with children, exposes them to harassment from every type of man. The way they can protect themselves is to share the shack with other people.
*For me it’s important, and it’s ok for me as a woman to live with men to take care of myself and be ok here, so that no one touches me or come near my shack, so that there are no women alone*. MW5

### 3.4. The Papers

For all the study participants, their main objective was to obtain “the papers”, to obtain legal residency in Spain. Due to the negative consequences of being in an “illegal” situation, they see legal regularization as a means to improve their lives.

#### 3.4.1. Consequences of Being “without Papers”

*We live without dignity, the papers give us dignity, without papers, you are not worth anything*. MW12

*Without papers, you can’t do anything, waiting to obtain the papers, and get by*. MW5


*Now I have a problem, and it is that when I look for work without papers, no one wants to give me work.*
MW12

The situation of irregular work for migrant individuals in the fields is exploited by farmers who can have workers without working rights and with salaries that are lower than what is legally set.
*I’ve always worked very hard, and I’ve worked more than people who had papers, and I’ve asked by boss for papers, but he did not want to obtain them, he wants those without papers because he pays less.*MW4
*Work without a contract is very hard, and you have to do it when you don’t have papers. They (bosses) do not pay well, they pay a very low salary, 30, 32–35€, more or less, for 8 h of work, 8 full hours without sitting for a minute, and it’s.*MW7
*Ok, you have to work to eat, to do things that need to be done, and to help the family.*MW5

#### 3.4.2. Difficulties of “Obtaining Papers”

The legal residency of migrants is complicated (as mentioned in the Introduction) and the requirements for obtaining papers are known, as stated by one of our informants.
*I want to have my papers, I need to be registered and I need a contract, I’m suffering a lot to obtain these papers, without them both, I can’t do anything.*MW4

For the MW who live in a shantytown, being able to register as residents is difficult, as the city council does not recognize the shacks as homes.
*I came to the city council to ask for the registration of residency, but they told me that I can’t ask for it because I live in a settlement. I want to register in my shack; I don’t have any other home.*MW13
*If the city council does not allow registration of residency in the settlement, it is a problem.*MW7

Not being able to register for residency in the shacks has led to the appearance of a lucrative business that takes advantage of this situation and increases the misery of the MW: the sale of residency registrations.
*Although you don’t live with the person, you pay, and he gets you the resident registration paper, and afterwards, he writes a report to ask for social integration…although you don’t have money, you buy it on credit, and you give it to the person who will write the report…even though you don’t have money…*MW1
*I bought my resident registration; 300 at first, and then 1500, almost two-thousand Euros to register, and soon I will have my residency papers*. MW5

The MW are aware of the importance of having resident registration for many aspects in their lives.
*The residency registration* (paper) *is not only important for the documents, but it’s also important for obtaining the healthcare card and for many things. It’s also important to have a photocopy in case the police stops you, to show that you’ve been here for a while and that you live here. It is important for many things, when you have residency, you can go to the consulate and obtain a passport, and all the Morocco things too.*
MW12

Another great difficulty in obtaining the resident registration papers is the need for a contract proposal, and there is also an “illegal market” that sells the contracts, which is known to everyone but tolerated.
*The work contract has to be paid as well; you look for a boss that could sell you a work contract for you, you talk to see how it can be done, if you have to pay everything up front or little by little, or if you can work with him and you pay, or if he doesn’t want you to work for him. More or less, the work contract costs between 5000 and 6000€.*MW5
*I didn’t even have money to eat, how am I supposed to pay for a contract that costs 5 or 6000€?*MW4
*Without a contract, without money, without anything, if you don’t have money, you can’t look for a contract, no one here gives a contract for free, this only happens with bosses with a person who has been there for a long time, two to three years, but in the end they always steal from you, but with people who work 1 or 2 months, they don’t give you a contract*. MW9

#### 3.4.3. Hope in the Papers

The MW place all their hopes and efforts in obtaining “the papers”. The papers would allow them to move on with their lives and live “a normal life”.
*We have to resist, try to work, and obtain papers to be able to leave, be able to rent, and live a normal life.*MW7
*Until we have the residency papers, and are able to move and see how we can live with a good job, to earn money and look for a house to rent, live peacefully, like everyone else*. MW5

Regularization is also seen as a means to be with their children, improve their living conditions, and to be able to go back to their country.
*…and to have a future for us and our children*. MW9
*When you have your papers, your work is more stable, in a warehouse or something like that, and afterwards, you will regroup with your family here in Spain, you can bring your parents or daughters.*MW6
*Hopefully I can finish my papers this year, so I can go back to my country to see my family, because I miss them a lot.*MW5

#### 3.4.4. False Hopes?

The only two MW we interviewed with legal residency in Spain and who still lived in the settlement mentioned that their regularization did not lead to a substantial improvement in their lives due to the illegal activities of the farmers.
*Yes, for unemployment too, but in my case, it’s the same, because I’ve already had two residency periods of 5 years, and I’ve never had unemployment, because even though you work, you don’t contribute* (to social security). *The put 3–4 days a month even though you work more…When will I contribute enough to be able to have unemployment?*
MW1
*I’ve worked for 400€ a month, and then I’ve worked for four euros an hour, and didn’t even have a full day of work, you can’t handle it.*MW3
*I worked for three months with a boss and he didn’t pay, even though I was registered and everything.*MW1

## 4. Discussion

In the present study, we have addressed the reality faced by MW who live in shantytowns within the context of the intensive agriculture in Huelva and Almeria, their migratory process, and their future expectations.

Most of the MW interviewed came to Spain via contracts from their countries of origin to work in the strawberry crops in Huelva, and the work conditions of exploitation they found, along with situations of bullying and sexual harassment by farmers, forced them to escape, thus becoming “illegals”, especially since 2018. Their testimonies match those from other studies [11,27]. This process of escape has led to the increase in the number of women in shantytowns in both Huelva and Almeria.

The results from our study show us that the lack of resources, the situation of irregular work, the lack of housing to rent, and the lack of owners willing to rent to migrants have led to what Arjona and Checa [33] denominate the condition of “impossible housing”. The difficulties in the access to rental properties or housing for seasonal workers have resulted in the growth of shantytowns due to the inaction of public institutions, as denounced in different reports from national and international organizations [10,34].

The living conditions in the shantytown settlements where these women live, with children on many occasions, among waste and sewage, and without potable water or legal access to electricity, was already denounced by a report from Philip Alston, a rapporteur on extreme poverty from the United Nations after his visit to Spain [34], affirming that the material conditions were the worst he had observed in any part of the world. It is a living situation that none of our informants imagined before starting their process of migration, and which exposes them and their children to important health-related problems [9].

Another problem in the settlements are the frequent fires in the shacks due to unsafe electrical installations and the need to heat the shack and cook inside. In the fires, the women lose the few belongings they have, which aggravates their situation of precariousness and their lack of resources, leading to permanent fear and anxiety [11,13].

The situation of the MW in the shantytown is especially difficult for those who are mothers; they send the little money they have to their children in Morocco, and the ones with children who live with them use the money to keep them in school and to guarantee that their basic needs are covered. Living far from their children, as well as having them in the settlements make them experience more pain and suffering.

The access to work for these MW is more complicated, especially because their situation of irregular work limits their work to the fields (where work is almost exclusively limited to the men). For the mothers, their children limit their mobility to find work. Our study brings to light the sexual harassment to which these women are subjected in the labor market; on some occasions, “the bosses” only hire them in exchange for sexual favors [11]. The lack of resources has pushed some women towards prostitution, which is also demonstrated in other studies [24,35].

The women who live alone in the settlements are exposed to different types of sexual violence and harassment [9,12,36], and to protect themselves, they must often share the shack with friends.

The administrative and judicial vulnerability, associated with a high labor precariousness, results in the living conditions of the migrants employed in the agricultural sector being extreme [5]. The MW explain their “life without dignity” as a result of their irregular situation in Spain, which paralyzes their migration project, and which condemns them to labor exploitation, to live in a settlement, and to not be able to go back to their country of origin to see their families.

On their road to regularization, to live a life “like everyone else”, they find great difficulties as a result of their own extreme situation. As explained above, the most common path towards regularization is via social or work ties, which depends on their resident registration in a Spanish municipality after three years and having a contract proposal. We found that the city halls where the settlements were located denied registering these women in the settlements where they live, despite the existence of laws that force them to do so [37]. This legislation specifically refers to the registration of inhabitants in substandard housing and the possibility of registering the people who live in shantytowns. Their non-registration in the shantytowns has resulted in the creation of an illegal business in which the people with housing offer registration in the form of an address in exchange for large sums of money that can reach EUR 1800. On the other hand, it is also frequent for farmers to sell work proposal contracts for EUR 6000–8000. The situations shown by our study are known by the public and the institutions; however, the complaints against those responsible are scarce.

This research study also interviewed MW with legal residency permits in Spain who live in settlements, and the work conditions in intensive agriculture in Huelva and Almeria, with their high instability and lack of compliance of work legislation, along with the absence of public policies of housing for field workers [11], make it impossible for them to improve their lives and abandon the settlements.

### Limitations and New Research Lines

We consider this small group of participants as both a strength and weakness of the present study. We recognize that we cannot generalize our findings beyond this group of MW. However, this study does not try to provide generalizations; instead, the objective was to present the reality of this group of women to shed light on their situation.

This study motivates us to further delve into the reality of these women and into the repercussions on the physical and mental health of the living conditions and the extreme vulnerability in which they survive.

## 5. Conclusions

The present study delves into the life experiences of a group of MW who live in shantytowns in areas of intensive agriculture in Southern Spain.

The increase in the number of MW and children in shantytowns in the last few years places a demand for specific programs destined to work with the women who live there, due to their greater vulnerability, which prioritize mothers who live with their children.

The inhumane and extreme living conditions in the settlements demand from society a search for urgent housing solutions for farm workers who live in them via the development of plans by the public administrations in charge and the local farm owners.

The difficulties in the regularization of these women limit their personal development and condemn them to live in the settlements. It is urgent to comply with the current legislation, to register them in the settlements where they live, and to seek out the illegal “selling” of residency registrations and contracts.

Given that the demand for workers for intensive agriculture in Southern Spain is continuous and rising, we propose the special regularization of farm workers “without papers” and a greater monitoring of the lack of compliance of the existing labor legislation in the field.

The living and working conditions of farm workers we found during the development of the present research study make us think about the society we are building in which the rights of those who are essential for producing our food are not recognized.

## Figures and Tables

**Table 1 ijerph-20-05524-t001:** Characteristics of the informants.

Code	Interview Language	Age	Number of Children	Country of Birth	Speaks Spanish	Arrival to Spain	Administrative Situation	Location
MW1	Arabic	41	2	Morocco	Yes	2007	Resident	Almeria
MW2	Arabic	41	2	Morocco	No	2018	Undocumented	Almeria
MW3	Arabic	35	1	Morocco	Some	2015	Resident	Almeria
MW4	Arabic	34	1	Morocco	Yes	2017	Undocumented	Huelva
MW5	Arabic	26	0	Morocco	Yes	2017	Undocumented	Almeria
MW6	Arabic	38	1	Morocco	Some	2019	Undocumented	Almeria
MW7	Arabic	28	0	Morocco	No	2019	Undocumented	Almeria
MW8	Arabic	29	0	Morocco	Some	2018	Undocumented	Huelva
MW9	Arabic	32	2	Morocco	No	2018	Undocumented	Huelva
MW10	Arabic	39	1	Morocco	No	2019	Undocumented	Huelva
MW11	Arabic	27	0	Morocco	Some	2018	Undocumented	Huelva
MW12	Arabic	30	1	Morocco	Some	2018	Undocumented	Huelva
MW13	Arabic	33	1	Morocco	Yes	2017	Undocumented	Huelva

**Table 2 ijerph-20-05524-t002:** Thematic categories.

Thematic Categories	Sub-Categories	Codes
Dreams vs. reality	Broken dreams	
	Reality	The arrival
		Work exploitation
Life in the settlement	Arriving at the settlement	Impossibility of renting a house
	Women in the settlement	Novelty
	Living conditions	Working and living under plastic
		Fires; permanent fear
Worse for women	Children	In Morocco; the sadness
		In Spain; desperation, limitation
	More difficulties for working	
	Without resources	
	Fear of harassment	
The papers	Consequences of no papers	Work exploitations
	Difficulties in obtaining the papers	Residency registration
		The contract
	Hope in the papers	Leaving the shantytown
		A normal life
		Improving their children’s lives
		Seeing the family

## Data Availability

The data presented in this study are available upon request from the corresponding author.

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
