# Peer review of "Migrant Women in Shantytowns in Southern Spain: A Qualitative Study"

_ijerph, 2023, doi:10.3390/ijerph20085524_

Round 1
Reviewer 1 Report
Migrant women in shantytowns in Southern Spain. A case- based study
This paper reports on the findings of a qualitative study looking into experiences of migration of female agricultural workers from Morocco living in shanty towns in the South of Spain, and their aspirations for the future. Recruitment aimed for purposeful sampling, to recruit women with diverse characteristics e.g. age, years in Spain, and Spanish literacy. The authors conducted semi-structured interviews with 13 women living in Huelva and Almeria. Women differed in their family and legal situation. Four thematic categories are being reported: Dreams vs Reality; Life in the settlement; Worse for women, and The papers. Findings reflect other research findings and reports which present the desperate situations experienced by migrant women working in either agricultural but also domestic work contexts, and the authors conclusions reflect numerous others found in the literature.
I found the manuscript difficult to read and it would benefit from significant editing to improve its grammar and intelligibility. Even though the authors present this as a case-based study, the methods described do not align with case-based study methodology and the authors do not define what their case is. The participants represented a diverse mix in terms of their social context i.e. where they lived, legal status, family situation. It would be more accurate to present this as a qualitative study, exploring the experiences of Moroccan women living in shantytowns the south of Spain. The authors need to provide more background context i.e. did women live in the same shantytowns in Huelva and Almeria? Where they linked in some way to each other?
It would be useful to see the interview topic guide to get an idea what participants talked about.
Was the interviewer one of the authors? If not, what were their characteristics e.g. student? What was her relationship to the research team?
It is not clear what analytical approach the researchers adopted in analysing the data e.g. is this a descriptive analysis? Clarke and Brown’s thematic analysis? Etc. The four thematic categories have considerable conceptual overlap between them (e.g. The code “work exploitations” appears under two categories, and the code “disillusionment” coming under “Life in the settlement” conceptually might relate better to Dreams vs Reality. Overall analysis appears to be very crude rather than in depth. The findings section resembles more a presentation of raw data rather than the outcome of a rigorous analytical process. There is no evidence that the analysis has gone beyond the basic descriptive to the conceptual. The findings reflect topics raised in the introduction rather novel and unique experiences and realities of the women participants, and this is also reflected in the discussion where the authors present findings and conclusions from other studies which reiterate what this paper presents, rather than adding to the existing literature. In the introduction the authors discuss the benefit of a gender approach to migrant research and allude to intersectionality, but these are not woven into their findings and discussion. Women are portrayed as lacking any agency and as victims and not enough emphasis is given to the ways they assert their agency to achieve their aspirations e.g. as strong negotiators or using the resources available to protect themselves. For example the following excerpt alludes to a woman who is aware of how to use men for protection in a pragmatic way, rather than as a victim being taken advantage of
For me it’s important, and it’s ok for me as a woman to live with men to take care of myself and be ok here, so that no one touches me or come near my shack, so that there are no women alone. MW5
I attach a copy of the PDF with more detailed comments.

Reviewer 2 Report
This is an interesting exploratory study of a new migration pattern in southern Spain, of women migrating to work on agricultural labor contracts and living in shantytowns when they do. The information provided in the article is very useful and informative. Several improvements can be made to improve data analysis and make the contribution to the literature stand out more significantly.
Overall, the introduction is well written and provides a good overview of this migratory pattern and why we need to know more about the experiences of migrant women in southern Spain. The rationale for the study is generally well explained. I recommend, however, some edits to the paragraph at the top of page three. The discussion of gender as a variable is a quantitative framing that is not appropriate for this paper. There are many qualitative studies of the lived experiences of women migrants, and it would be better to couch this study more prominently in that body of literature. I also, then, recommend some additions to this section from the very robust literature on female migration to suggest that including women's perspectives is important as doing so offers insight into the contours of migratory patterns more broadly, and how inequality is perpetuated through temporary migration processes specifically.
The Materials and methods section also uses language in places that seems inappropriate and does not highlight the strengths of the data. This is a small sample size of 13 individuals, this is fine; however, reaching saturation in such a small sample is not really feasible - especially with relatively short 50-minute interviews in two different sites. It would be better to describe the sample as one in which NGOs provided recommendations to key informants in the community. In this way, these voices are representative. It is also important in a qualitative study such as this to describe the roles of each of the authors in the study and their connection to the research sites. Also, what year and month were the interviews collected? This matters in terms of time since arrival and in terms of the seasonal work performed. The analysis approach seems to be a more quantitative style and strategy of analyzing qualitative data. It would be good to be clear about the epistemological approach as it seems to contradict other statements in the manuscript about the value of assessing lived experiences of migration through a small-sample size interview process.
The results section is where I recommend the authors do the most revision. The style of presenting quotes without any interpretation undermines the quality of the analysis and effectively hides the most interesting findings of the data. It would be better to present the evidence from the interviews along with interpretation of the evidence. Then the discussion can focus instead on more synthesis across themes as it relates to patterns of female migration globally. In other words. I would integrate the current results and discussions sections and then rewrite the discussion section to be more focused on how these findings compare to experiences of female migrants - and specifically women in seasonal labor contracts - elsewhere.
Finally, I recommend the conclusion address some more specific policy interventions and outcomes based on this study. as written, the suggestions are broad in stroke and could use some more detail and explanation.
